# High Resistance to Antibiotics Recommended in Standard Treatment Guidelines in Ghana: A Cross-Sectional Study of Antimicrobial Resistance Patterns in Patients with Urinary Tract Infections between 2017–2021

**DOI:** 10.3390/ijerph192416556

**Published:** 2022-12-09

**Authors:** Benjamin Asamoah, Appiah-Korang Labi, Himanshu A. Gupte, Hayk Davtyan, Georgette Marfo Peprah, Forster Adu-Gyan, Divya Nair, Karlos Muradyan, Nasreen S. Jessani, Paul Sekyere-Nyantakyi

**Affiliations:** 1MDS-Lancet Laboratories, Accra P.O. Box AC 533, Ghana; 2World Health Organization, Country Office, Accra P.O. Box MB 142, Ghana; 3Narotam Sekhsaria Foundation, Mumbai 4000 021, India; 4Tuberculosis Research and Prevention Center, Yerevan 0014, Armenia; 5International Union Against TB and Lung Disease (The Union), 75006 Paris, France; 6Centre for Evidence based Health Care, Department of Global Health, Stellenbosch University, Cape Town 800, South Africa; 7Department of International Health, Johns Hopkins Bloomberg School of Public Health, Baltimore, MD 21205, USA

**Keywords:** urinary tract infection, Ghana, antimicrobial resistance (AMR), multi-drug resistance (MDR), MDS Lancet Laboratories, Access, Watch and Reserve (AWaRE) classification, uropathogens

## Abstract

Management of urinary tract infections is challenged by increasing antimicrobial resistance (AMR) worldwide. In this study, we describe the trends in antimicrobial resistance of uropathogens isolated from the largest private sector laboratory in Ghana over a five-year period. We reviewed positive urine cultures at the MDS Lancet Laboratories from 2017 to 2021. The proportions of uropathogens with antimicrobial resistance to oral and parenteral antimicrobials recommended by the Ghana standard treatment guidelines were determined. The proportion of multi-drug resistant isolates, ESBL and carbapenemase-producing phenotypes were determined. Of 94,134 urine specimens submitted for culture, 20,010 (22.1%) were culture positive. *Enterobacterales* was the most common group of organisms, *E. coli* (70.6%) being the most common isolate and *Enterococcus* spp. the most common gram-positive (1.3%) organisms. Among oral antimicrobials, the highest resistance was observed to ciprofloxacin (62.3%) and cefuroxime (60.2%) and the least resistance to fosfomycin (1.9%). The least resistance among parenteral antimicrobials was to meropenem (0.3%). The highest multi-drug resistance levels were observed among *Klebsiella* spp. (68.6%) and *E. coli* (64.0%). Extended-spectrum beta-lactamase (ESBL) positivity was highest in *Klebsiella* spp. (58.6%) and *E. coli* (50.0%). There may be a need to review the Ghana standard treatment guidelines to reflect increased resistance among uropathogens to recommended antimicrobials.

## 1. Introduction

Antimicrobial resistance (AMR) threatens human health by limiting the options for treating infection [1]. AMR has been declared as one of the top 10 public health threats to humanity by the World Health Organization and a direct obstacle to achieving the Sustainable Development Goals [1]. Illnesses resulting from organisms with AMR lead to long hospital stays and require broad-spectrum and expensive antimicrobials for treatment, resulting in large financial burdens for patients [1].

Urinary tract infections (UTIs) are one of the leading causes of morbidity and a growing health care expenditure globally affecting 150 million people each year [2,3]. Morbidity and mortality from UTIs are higher in lower middle-income countries (LMICs) than in higher-income countries [4]. UTIs are most commonly caused by uropathogenic *Escherichia coli* (UPEC), *Klebsiella pneumoniae* and *Enterococcus faecalis* [5]. 

Widespread and indiscriminate use of antibiotics has led to the development of an alarming level of AMR among uropathogens, including the emergence of extended-spectrum beta-lactamase (ESBL) producing and carbapenemase-producing organisms [6]. ESBL-producing and carbapenemase-producing organisms are typically multi-drug resistant [6]. This mandates the use of the Watch and Reserve groups of antibiotics from the AWaRe (Access, Watch, Reserve) classification of the WHO [7]; these antibiotics are typically parenteral and may require hospitalization for the purposes of administration [8]. Common risk factors associated with resistant UTIs are urinary catheterization, previous hospitalization, previous antibiotic use and residence in a nursing home facility [9].

Although the burden of UTIs in Africa is not clearly documented, prevalence rates ranging between 10.1% and 76.6% have been reported in different sub-populations from across the continent [9,10,11]. In the same studies, AMR rates to antibiotics used for treating UTIs as high as 95% have been reported [9,10,11]. Whilst there is limited information about the general national prevalence of UTIs in Ghana, two studies have reported 42.75% prevalence in pregnant women [12] and 86% among hospitalized adults [13]. The proportion of multi-drug resistant (MDR) UTIs has been reported to be as high as 93.6% in Ghana [10,14]. AMR is a recognized threat in Ghana, and there is a national policy and action plan based on the global action plan on AMR [15]. Ghana is currently amongst the countries that are reporting to the global AMR surveillance system (GLASS) starting from 2021 [16].

A nation-wide study exploring AMR patterns in various uropathogens across several years and in different regions of Ghana could strengthen the knowledge and evidence base to guide policy and practice in line with the five objectives (Awareness, Surveillance, Infection prevention and control, Antimicrobial usage and Research and innovation) of the global action plan on AMR by the World Health Organization [16].

MDS Lancet Laboratories Gh. LTD is a member of the Cerba Lancet Africa group, a leading network of clinical pathology and medical diagnostics in Africa. MDS Lancet Laboratories Gh. LTD is currently the largest private pathology laboratory in Ghana, providing service to healthcare facilities in the private and public sectors.

In this study, we aimed to identify the AMR patterns and trends of MDR among uropathogens isolated from the urine samples of patients with suspected UTIs processed at MDS Lancet Laboratories, Ghana, over a five-year period from 2017 to 2021. Specifically, we sought to determine trends in (i) the yield of culture testing and proportions of all identified species, (ii) the resistance patterns of antibiotics grouped by route of administration and the WHO AwaRe classification, (iii) the factors associated with AMR and (iv) *E.coli* antibiotic resistance over the study period.

## 2. Materials and Methods

### 2.1. Study Design

This was a cross-sectional study involving the secondary analysis of laboratory data of urine cultures and antimicrobial susceptibility testing conducted at the MDS Lancet Laboratories, Ghana, between 2017 and 2021.

### 2.2. Study Setting

#### 2.2.1. General Setting

Ghana is classified as a low middle-income country (LMIC) in West Africa with a total population of approximately 30.8 million. Accra is the capital city located at the coastal belt and has a total population of approximately 5 million [17]. Health care in Ghana is mainly provided by public sector facilities but with significant contribution from private sector facilities including laboratories. The public healthcare system consists of tertiary care hospitals, district (secondary) hospitals and primary health care facilities [18]. According to the law, patients need prescriptions from a medical doctor to get antibiotics [15]. However, it is possible to get antibiotics without prescriptions from private pharmacies. The Ghana Standard Treatment Guidelines (2017) suggest the following antibiotics for the treatment of UTIs: ciprofloxacin, cefuroxime and amoxicillin/clavulanic acid for uncomplicated infections; and gentamicin and ceftriaxone for complicated infections [19].

#### 2.2.2. Specific Setting

MDS Lancet Laboratories Gh. LTD is a private medical laboratory with 29 locations dotted throughout 10 of the 16 administrative regions of Ghana. The laboratory processes approximately 148,500 samples per month and receives more than 50,000 urine (routine examination and cultures) samples per year. All testing and information management processes are conducted under strict quality control mechanisms as defined in the laboratory’s standard operating procedures. The laboratory holds ISO 15189 2012 certification. The headquarters/main laboratory is located in Accra. The microbiology department where this research was carried out has standard quality assurance mechanisms in place at the pre-analytical, analytical and post-analytical stages.

#### 2.2.3. Sample Collection and Processing

The urine samples for this study originated from patients diagnosed with suspected UTIs in health care facilities (hospitals and clinics) in both the public and private sectors. Urine samples are usually transported by the patients or their relatives in sterile urine containers. All microbiology samples from the different branches that are meant for culture are transported to the microbiology laboratory in Accra within 24 h for processing using temperature-controlled, sample-transportation bags.

All urine samples received in the laboratory are registered and plated on cysteine lactose electrolyte deficient (CLED) agar, McConkey’s agar, blood agar and antimicrobial plates [20]. The first three media are used to determine the presence of microorganisms in the sample, while the latter is used to determine the presence of antibiotics in the urine and provides information on the use of antibiotics prior to sample collection. After 24 h of incubation, cultures are assessed and positive samples are further processed for identification of the specific microorganisms by performing indole and oxidase and other biochemical tests as well as the Analytical Profile Index (API) or Microscan [21]. Antimicrobial susceptibility testing is performed using the Kirby–Bauer disc diffusion method on Mueller–Hinton agar [22]. Zones of inhibition are interpreted using the Clinical and Laboratory Standards Institute (CLSI) guidelines [23].

### 2.3. Study Population

The study population included all urine samples submitted for culture and susceptibility testing at the MDS Lancet Laboratories in Ghana from January 2017 to December 2021.

### 2.4. Data Variables

For this study, the following variables were extracted from the laboratory database and used in the analysis: Patient ID and demographic characteristics, sample collection date and results entry date, the isolated organisms, ESBL status (for gram negatives) and antibiotic susceptibility patterns.

### 2.5. Sources of Data, Data Collection and Validation

The MDS Lancet Laboratories uses an electronic laboratory information system (LIS) where all the data related to the patients and samples are entered. Right from the collection of the sample, information is entered into the system and verified at each stage from transportation to processing of the samples. The results of the tests are reported in the same LIS, and the information becomes readily available for collection at any MDS Lancet branch. The LIS of the MDS Lancet Laboratories was used to extract the data for the study period. The data was extracted into MS Excel format and analyzed using STATA^®^ (version 16.0 Copyright 1985–2019 StataCorp LLC, College Station, TX, USA). 

### 2.6. Statistical Analysis

Frequency and proportions were used to summarize the demographic and microbiological characteristics. MDR was defined as resistance to at least one antimicrobial agent from three or more classes of antibiotics [24]. The presence of resistance to any antibiotic and MDR among all culture-positive samples were summarized as proportions with a 95% confidence interval (95% CI). Resistance to commonly prescribed antibiotics (individually and grouped according to the AWaRe classification) for different isolates was reported as a percentage. To identify the factors independently associated with a UTI due to MDR organisms in a patient, a modified Poisson regression with robust variance estimator was carried out. A patient was considered to have a UTI due to an MDR organism if any of the isolates from their urine sample turned out to be MDR. As this was a cross-sectional study, prevalence ratios (PR) and adjusted prevalence ratios (aPR) were reported as measures of association. The level of significance for all statistical tests was set at a *p*-value of 0.05 [25].

## 3. Results

### 3.1. Patient Characteristics and Proportions of All Identified Bacterial Isolates in Urine Samples

Over the five years, a total of 328,073 urine samples were submitted for processing. Of these, 94,134 samples were submitted for culture and susceptibility testing at the MDS Lancet Laboratories, and 20,010 (22.1%) yielded at least one organism on culture. Among the culture-positive samples, 793 (4%) yielded more than one organism. Table 1 shows the characteristics of the patients with culture-positive urine samples. Most culture-positive samples (38.8%) were from patients aged 15–44 years. Positive urine samples were more common in females (72.5%) compared to males, and the majority of positive urine samples were obtained from the Greater Accra region (63.5%).

Antibiotic residue was present in 1986 (9.9%) culture-positive samples. Isolates resistant to at least one antibiotic were found in 65.3% (95% CI: 64.6–66.0%) of cultures, and multi-drug resistant isolates were found in 63% (95% CI: 62.3–63.6%) of cultures.

### 3.2. Uropathogen Characteristics

Table 2 shows the distribution of uropathogens from the urine cultures. *Enterobacterales* was the most common group of organisms with *Escherichia coli* (70.6%), *Klebsiella* spp. (15.0%) and *Proteus* spp. (5.4%) being the most common, while *Enterococcus* spp. (1.3%) were the most common gram-positive organisms isolated. This pattern of organism distribution was similar across all age groups, region of sample origin and year of specimen collection. The full list of isolated pathogens and patient characteristics are provided in Appendix A.

### 3.3. Trends in Urinary Bacterial Isolates over Five Years, 2017–2021

Annual trends in urinary bacterial isolates for the top five GNBs and gram-positive organisms over five years (2017–2021) are shown in Figure 1.

The total number of isolates over the five years per year are as follows: 2017 (1568), 2018 (2456), 2019 (4193), 2020 (5050) and 2021 (7543).

Figure 1 shows a general increase in the number of isolates for the top five GNBs over the study period.

### 3.4. Resistance Patterns of Antibiotics Grouped by Route of Administration and the WHO AWaRe Classification

Table 3 and Table 4 show the proportion of the most common uropathogens resistant to oral antibiotics as recommended by the national standard treatment guidelines including fosfomycin and common parenteral antibiotics that were likely to be used as second-line agents for treatment. Among *Enterobacterales*, the least resistance was observed to fosfomycin—*E. coli* (2.9%), *Klebsiella* spp. (9.9%) and *Proteus* spp. (14.0%); the highest resistance was observed to ciprofloxacin (62.3%) and cefuroxime (60.2%). Among *Enterococcus* spp., the least resistance was observed to nitrofurantoin (1.5%).

The least resistance to parenteral antibiotics among *Enterobacterales* was to meropenem (0.3%) and tigecycline (1.7%). Among isolated *Enterococcus* spp., no resistance was observed to vancomycin. Among ESBL-positive organisms, the least resistance observed among oral antibiotics was to fosfomycin (5.4%) and nitrofurantoin (41.6%). Among the parenteral antibiotics, the least resistance was observed to amikacin (5.3%) and meropenem (1.2%). The details of the antimicrobial resistance patterns among ESBL-positive organisms are provided in Appendix A.

Resistance of *E. coli* to oral antibiotics (amoxicillin/clavulanic acid, cefuroxime, ciprofloxacin, fosfomycin and nitrofurantoin) was constant over the study period as shown in Figure 2.

### 3.5. Multi-Drug Resistance and ESBL Positivity

The proportion of isolates that showed MDR was highest in *Klebsiella* spp. (68.6%) and *E. coli* (64.0%). Among *Enterobacterales*, ESBL positivity was highest among *Klebsiella* spp. (58.6%) followed by *E. coli* (50.0%) and least in *Proteus* spp. (5.3%). In addition, the proportions of ESBL were stable across the study years. The details of the trends in MDR and ESBL positivity among commonly isolated uropathogens are provided in Table 5. Gram-positive bacteria are not presented due to the low number of resistant isolates.

### 3.6. Factors Associated with Antimicrobial Resistance

Age, gender, specimen location and presence of antibiotic residue in urine were found to be independently associated with a patient having a MDR uropathogen (Table 6). The patients aged 65 years and above (aPR 1.16, 95% CI: 1.11–1.22) and males (aPR: 1.13, 95% CI: 1.11–1.16) had a significantly higher risk of a UTI due to a multi-drug resistant uropathogen. The patients with antibiotic residues in their urine (aPR 1.40, 95% CI: 1.37–1.43) had a significantly higher risk of a UTI due to a multi-drug resistant uropathogen.

## 4. Discussion

In this study, the culture yield over the study period was 22.1%. This is similar to results obtained in other studies in which the positivity ranged from 17% to 37% [26,27]. *Enterobacterales* was the most common group of pathogens isolated with *E. coli* and *Klebsiella* spp. being the most common isolates, and *Enterococcus* spp. being the most common gram-positive organisms. These organisms predominate the human gut flora, which is known to be the common source of autoinfection for UTIs. These findings are similar to the results in other studies where *E. coli* at 68.3% and *Klebsiella pneumoniae* at 31.7% were the most predominant isolates from UTIs [28]. In another study from India, *E.coli* was the predominant isolate at 77.9% followed by *Klebsiella* spp. at 22.1% [29].

High levels of resistance to the Standard Treatment Guidelines (STG) recommended antibiotics for treating urinary tract infections were observed in this study. This situation was more acute among *Enterobacterales*. For example, among *E. coli* isolates, resistance to cefuroxime and ciprofloxacin was 60.2% and 62.3%, respectively. This situation can result in poor treatment outcomes, including prolonged treatment for empirically treated UTIs. The oral antibiotics with the least resistance in this study were nitrofurantoin and fosfomycin. Fosfomycin is a new agent on the Ghanaian market and is not widely available. Thus, its use is restricted, and it was expected that resistance to this antibiotic would be low. Although nitrofurantoin is an old drug, the side effects arising from its use may have resulted in its limited use and, consequently, the relatively low resistance observed compared to the other agents [30,31].

In a recent review of AMR among uropathogens in the Asia–Pacific region, a similar pattern was observed; resistance to commonly used drugs, such as ciprofloxacin, sulfamethoxazole–trimethoprim and ceftriaxone, ranged from 33% to 90% with low resistance observed to nitrofurantoin from 2.7% to 34% and to fosfomycin from 1.7% to 1.8% [32].

The resistance to parenteral antibiotics, such as ceftriaxone and gentamicin that are recommended for the treatment of complicated UTIs, was also high. This may be due to the relatively high prevalence of ESBL observed among *Enterobacterales*. ESBL genes are typically borne on plasmids, which are known to carry multiple resistance genes making affected organisms multi-drug resistant [33]. The proportion of ESBL among *Enterobacterales* in this study was high, similar to findings from other studies from Ghana [14].

The proportion of ESBL among organisms seemed be constant over the five years under consideration. Our study suggests that the agents useful against ESBL-producing organisms are fosfomycin, meropenem and amikacin. These agents may be considered for the empirical treatment of a UTI when risk factors for MDR infection exist, such as previous hospitalizations or antibiotic use. MDR was a common characteristic among the majority of uropathogens isolated. This suggests AMR may be a major problem in Ghana.

Among the different age groups, the risk of having a MDR infection was associated with older patients. Increasing age is associated with an increased use and misuse of antibiotics, a situation likely to increase MDR risk. In other studies, increasing age was found to be associated with an increased risk of having a MDR UTI and, hence, an increased use of antibiotics [34,35]. A study in the United Kingdom (UK) showed an increased risk of MDR with age, with odds ratios of 1.8, 2.69, 3.22 and 3.62 for the age ranges 50–59, 60–69, 70–79 and >80 years, respectively [35].

Whilst the prevalence of UTIs was common in women in this study, the risk of MDR infection was significantly higher in males compared to women. UTIs in males are usually associated with a urological abnormality, such as prostatic hyperplasia or posterior urethral valves that are usually associated with instrumentation and risk of infection [36,37,38]. Lastly, the highest risk of MDR was found in patients with the presence of antibiotics in the urine. This presupposes that most of these patients had difficult-to-treat infections and were probably started on ineffective antibiotics or had misused antibiotics, all of which may have contributed to the development of resistance.

A major strength of this study is the large number of isolates included from different regions of Ghana, and the study represents one of the largest datasets on AMR in uropathogens on the African continent as well as in Ghana. This data may give a better understanding of AMR in uropathogens in Ghana, although the preponderance of data from the Greater Accra and Ashanti regions may introduce some bias into data representativeness. Another strength of this study is that we followed STROBE (Strengthening the Reporting of Observational Studies in Epidemiology) guidelines to report the findings of our study [39].

A major limitation of this study, however, is the inability to segregate the data into outpatient and inpatient because of data unavailability. Such data would have allowed for the segregation of community-acquired and hospital-acquired infections and their association with AMR. This study, being a laboratory-only based surveillance, has the potential to overestimate the level of resistance among uropathogens as has been shown in other studies [40].

This study has significant implications for policy and practice. Based on the current data, there may be a need to revise the STG guidelines on antibiotics to be used for treating UTIs, especially complicated infections, to reflect the large proportions of AMR observed among uropathogens. As these guideline changes are considered, there will be a need to implement antibiotic stewardship measures to prolong the usefulness of agents, such as fosfomycin, nitrofurantoin, amikacin and meropenem that may be recommended as alternate antibiotics. Also, there will be a need to investigate the utility of other oral antibiotics for treating UTIs to offer clinicians and patients more therapeutic options in the face of growing MDR infections. In the long term, there is a need to establish laboratory-based surveillance of uropathogens nationwide augmented by large population-based studies covering large geographical regions to inform policy and practice. This is important to correct for the selection bias of laboratory surveillance for MDR organisms. Also, clinicians should be encouraged to routinely utilize culture and susceptibility testing during the management of UTIs to avoid poor treatment outcomes because of resistant strains.

## 5. Conclusions

Over the five-year period, gram-negative pathogens, especially *E. coli*, were the most common cause of UTIs. High levels of resistance to STG recommended oral and parenteral antibiotics for the management of UTIs were observed. There may be a need to review the current STG guidelines to reflect the high numbers of AMR in the most common isolates.

## Figures and Tables

**Figure 1 ijerph-19-16556-f001:**
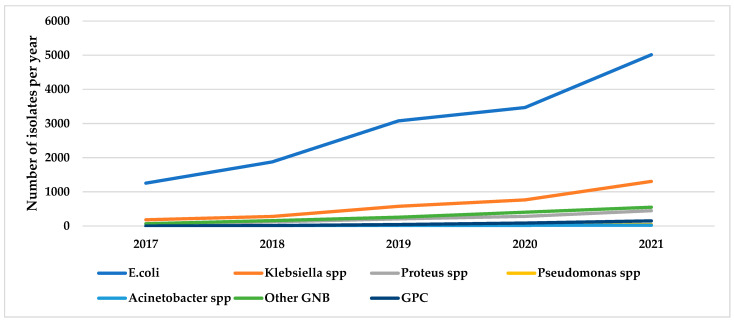
Annual trends in the number of isolates for the top five GNBs and gram positives from urine samples of patients with urinary tract infections at MDS Lancet laboratories, Ghana, from 2017 to 2021. Number for each year is the total of each isolate for the particular year; Other GNB: all gram-negative bacilli isolated in the laboratory apart from *E.coli*, *Klebsiella* spp., *Proteus* spp., *Pseudomonas* spp. and *Acinetobacter* spp.; GPC: Gram-positive cocci.

**Figure 2 ijerph-19-16556-f002:**
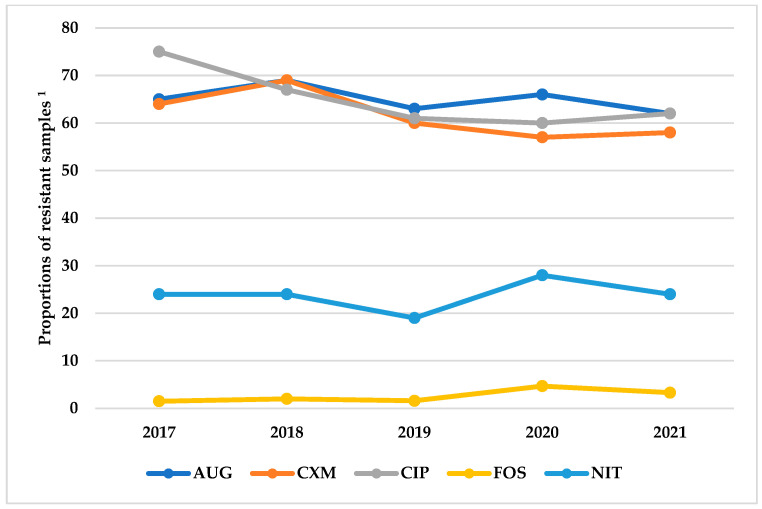
Annual trends in the proportions of resistance to selected antimicrobials among *E. coli* isolated from urine samples of patients with urinary tract infections at MDS Lancet Laboratories, Ghana, from 2017 to 2021. ^1^ Proportion for each year calculated with the number of samples in which *E. coli* was isolated in the year as the denominator (2017 = 1256, 2018 = 1877, 2019 = 3078, 2020 = 3469, 2021 = 5015); AUG: Amoxicillin/clavulanic acid; CXM: Cefuroxime; CIP: Ciprofloxacin; FOS: Fosfomycin; NIT: Nitrofurantoin; ESBL: Extended-spectrum beta-lactamase.

**Table 1 ijerph-19-16556-t001:** Socio-demographic and microbiological characteristics of patients with urinary tract infections with culture-positive samples ^1^ analyzed at MDS Lancet Laboratories, Ghana, from 2017 to 2021.

Characteristics		*n*	(%)
Total number of patients		20,010	(100)
Age in years			
	<15	1474	(7.4)
	15–44	7771	(38.8)
	45–64	4539	22.7
	≥65	6040	(30.2)
	Unknown	186	0.9
Sex			
	Male	5484	(27.4)
	Female	14,505	(72.5)
	Unknown	21	(0.1)
Geographic location of urine specimens			
	Accra	12,702	(63.5)
	Ashanti	4.266	(21.3)
	Others	3042	(15.2)
Antimicrobial substance in urine ^2^	Present	1986	(9.9)
Number of bacterial isolates identified			
	One	19,217	(96.0)
	Two	786	(3.9)
	Three	7	(<0.1)
Antimicrobial resistance			
	Resistance to at least one antimicrobial	13,079	(65.3)
	MDR ^3^ in at least one isolate	12,609	(63.0)

^1^ Culture-positive samples mean the urine sample yielded at least one bacterial isolate; ^2^ Presence of antimicrobial substance in urine sample implies that patients were on antibiotics before their urine sample was taken; ^3^ MDR—Multi-drug resistance defined as resistance to three or more antibiotic drug classes.

**Table 2 ijerph-19-16556-t002:** Distribution of common uropathogens isolated from the urine samples of patients with urinary tract infections analyzed at MDS Lancet Laboratories, Ghana, from 2017 to 2021 based on age and region.

	Gram-Negative Bacteria (GNB)												Gram-Positive Bacteria (GPC)	
	*E. coli*		*Klebsiella* spp.		*Proteus* spp.		*Pseudomonas* spp.		*Acinetobacter* spp.		Other GNB			
	*n* (%) ^1^		*n* (%) ^1^		*n* (%) ^1^		*n* (%) ^1^		*n* (%) ^1^		*n* (%) ^1^		*n* (%) ^2^	
**Age groups in years**														
<15	997	(6.8)	272	(8.7)	104	(9.3)	5	(5.0)	9	(18.8)	143	(9.9)	26	(8.8)
15–44	5887	(40.1)	1141	(36.7)	374	(33.4)	15	(15.0)	17	(35.4)	501	(34.8)	137	(46.4)
45–64	3329	(22.7)	735	(23.6)	237	(21.2)	21	(21.0)	4	(8.3)	309	(21.5)	64	(21.7)
≥65	4341	(29.5)	942	(30.3)	395	(35.3)	59	(59.0)	18	(37.5)	472	(32.8)	65	(22.0)
**Regions**														
Accra	9437	(64.2)	1921	(61.7)	723	(64.6)	65	(65.0)	27	(56.3)	824	(57.2)	186	(63.1)
Ashanti	3080	(21.0)	659	(21.2)	280	(25.0)	18	(18.0)	6	(12.5)	348	(24.2)	77	(26.1)
Others	2178	(14.8)	532	(17.1)	117	(10.4)	17	(17.0)	15	(31.3)	268	(18.6)	32	(10.8)
Total	14695	(100)	3112	(100)	1120	(100)	100	(100)	48	(100)	1440	(100)	295	(100)

^1^ Percentages reported out of all culture-positive samples with gram-negative bacilli (GNB) isolates reported for each category; ^2^ Percentages reported out of all culture-positive samples with gram-positive cocci (GPC) isolates reported for each category.

**Table 3 ijerph-19-16556-t003:** Prevalence of antimicrobial resistance to common oral antibiotics used in treating UTIs in Ghana among the top six GNBs and *Enterococcus* spp. isolated from urine samples of patients with UTIs at MDS Lancet Laboratories, Ghana, from 2017 to 2021.

	Ampicillin	Amoxicillin Clavulanate	Nitrofurantoin	Ciprofloxacin	Cefuroxime	Fosfomycin
	*n* (%) ^1^	*n* (%) ^1^	*n* (%) ^1^	*n* (%) ^1^	*n* (%) ^1^	*n* (%) ^1^
*E. coli*	N/A	9411 (64.0)	3575 (24.3)	9161 (62.3)	8851 (60.2)	435 (2.9)
*Klebsiella* spp.	N/A	2083 (66.9)	1639 (52.7)	1659 (53.3)	1986 (63.8)	308 (9.9)
*Proteus* spp.	N/A	105 (9.4)	N/A	151 (13.5)	91 (8.13)	6 (14.0)
*Acinetobacter* spp.	N/A	N/A	N/A	9 (18.8)	N/A	N/A
*Pseudomonas* spp.	N/A	N/A	N/A	41 (41)	N/A	N/A
*Enterococcus faecalis*	9 (3.3)	N/A	4 (1.5)	N/A	N/A	5 (1.9)

^1^ Percentages calculated with the total number of urine samples that yielded the particular species as the denominator. Antibiotics in Green belong to the Access group, and those in Yellow belong to the Watch group as per the WHO AWaRe classification 2021; UTI—urinary tract infection; GNB—gram-negative bacilli; GPC—gram-positive cocci; N/A—Not applicable (not tested for the isolate).

**Table 4 ijerph-19-16556-t004:** Prevalence of antimicrobial resistance to common parenteral antibiotics used in treating UTIs in Ghana among the top uropathogens isolated from urine samples of patients with UTIs at MDS Lancet Laboratories, Ghana, from 2017 to 2021.

Organism	Ak	Caz	Cro	Mem	Piptaz	Tige
	*n* (%) ^1^	*n* (%) ^1^	*n* (%) ^1^	*n* (%) ^1^	*n* (%) ^1^	*n* (%) ^1^
*E. coli*	461 (3.1)	7147 (48.62)	7362 (50.1)	42 (0.3)	7148 (48.6)	252 (1.7)
*Klebsiella* spp.	116 (3.7)	1802 (57.9)	1834 (58.9)	56 (1.8)	1803 (57.9)	173 (5.6)
*Proteus* spp.	3 (0.3)	46 (4.1)	46 (4.1)	3 (0.3)	17 (39.5)	N/A
*Acinetobacter* spp.	2 (4.2)	17 (35.4)	N/A	1 (2.1)	14 (29.2)	N/A
*Pseudomonas* spp.	21 (21)	30 (30)	N/A	19 (19)	27 (27.0)	N/A

^1^ Percentages calculated with the total number of urine samples that yielded the particular species as the denominator. Antibiotics in Green belong to the Access group, those in Yellow belong to the Watch group, and those in Red belong to the Reserve Group as per the WHO AWaRe classification 2021. Cro: Ceftriaxone; Cp: Cefepime; Piptaz: Piperacillin tazobactam; Caz: Ceftazidime; Mem: Meropenem; Ak: Amikacin; Vanc: Vancomycin; Tige: Tigecycline; UTI—urinary tract infection; GNB—gram-negative bacilli; N/A—not applicable.

**Table 5 ijerph-19-16556-t005:** Prevalence of multi-drug resistance (MDR) and extended-spectrum beta-lactamase (ESBL) positivity among commonly isolated bacteria causing UTIs in patients at MDS Lancet Laboratories, Ghana, from 2017 to 2021.

Organism	MDR						ESBL					
	2017	2018	2019	2020	2021	Overall	2017	2018	2019	2020	2021	Overall
	*n* (%) ^1^	*n* (%) ^1^	*n* (%) ^1^	*n* (%) ^1^	*n* (%) ^1^	*n* (%) ^1^	*n* (%) ^2^	*n* (%) ^2^	*n* (%) ^2^	*n* (%) ^2^	*n* (%) ^2^	*n* (%) ^2^
Gram negatives												
*E. coli*	871 (69.4)	1309 (69.7)	1958 (63.6)	2182 (62.9)	3087 (61.6)	9407 (64.0)	600 (47.8)	975 (51.9)	1585 (51.5)	1692 (48.8)	2492 (49.7)	7344 (50.0)
*Klebsiella* spp.	137 (74.5)	216 (77.4)	376 (65.2)	522 (68.3)	885 (67.7)	2136 (68.6)	106 (57.6)	181 (64.9)	329 (57.0)	441 (57.7)	765 (58.5)	1822 (58.6)
*Proteus* spp.	4(6.8)	14 (11.6)	33 (15.8)	32 (11.4)	59 (13.1)	142 (12.68)	4 (6.8)	6 (5.0)	16 (7.7)	11 (3.9)	22 (4.9)	59 (5.3)
*Pseudomonas* spp.	0 (0.0)	1 (16.7)	9 (40.9)	14 (51.8)	16 (35.6)	40 (40)	N/A	N/A	N/A	N/A	N/A	N/A
*Acinetobacter* spp.	0 (0.0)	0 (0)	0 (0.0)	6 (31.6)	4 (18.2)	10 (20.8)	N/A	N/A	N/A	N/A	N/A	N/A
Other Gram-negative rods	65 (94.3)	140 (89.1)	220 (85.3)	348 (86.1)	425 (77.0)	1198 (83.2)	44 (63.8)	90 (57.3)	131 (50.8)	126 (31.2)	185 (33.5)	576 (40)

^1^ Percentage of MDR isolates for each bacterial species is calculated with the number of samples that yielded the particular species as the denominator; ^2^ Percentages of ESBL-producing isolates for each bacterial species is calculated with the number of samples that yielded the particular species as the denominator; N/A = Not applicable.

**Table 6 ijerph-19-16556-t006:** Factors associated with multi-drug resistance in patients with urinary tract infections with culture-positive samples tested at MDS Lancet Laboratories, Ghana, from 2017 to 2021.

Factors	Total *n* ^1^	MDR *n* (%) ^2^	PR (95% CI)	aPR (95% CI)	*p*-Value
**Age in years**					
<15	1474	854 (57.9)	Ref	Ref	
15–44	7771	4595 (59.1)	1.01 (0.97–1.06)	1.05 (0.99–1.09)	0.06
45–64	4539	3020 (66.5)	1.14 (1.09–1.20)	1.14 (1.08–1.19)	<0.001
≥5	6040	4207 (69.6)	1.20 (1.14–1.25)	1.16 (1.11–1.22)	<0.001
Sex					
Male	5484	3989 (72.7)	1.20 (1.17–1.22)	1.13 (1.11–1.16)	<0.001
Female	14,505	8783 (60.5)	Ref	Ref	
**Specimen location ^3^**					
Accra	12,702	7931 (62.4)	Ref	Ref	
Ashanti	4266	2722 (63.8)	1.02 (0.99–1.05)	1.04 (1.01–1.07)	0.002
Others	3042	2131 (63.8)	1.12 (1.09–1.15)	1.09 (1.07–1.13)	<0.001
**Antimicrobial substance in urine**					
Present	1986	1742 (87.1)	1.43 (1.40–1.46)	1.40 (1.37–1.43)	<0.001
Absent	18,007	11,030 (61.2)	Ref	Ref	

^1^ Missing values in age (186), sex (21) and antimicrobial substance in urine (17) dropped from analysis; ^2^ Row percentages; ^3^ Specimen locations are the regions from where the samples were collected. Samples from regions outside greater Accra were well-packaged and transported in temperature-regulated specimen bags; MDR—Multi-drug resistance; PR—prevalence ratio; CI—confidence interval; aPR—adjusted prevalence ratio.

## Data Availability

The data presented in this study are available on request from the corresponding author.

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
