# Peer review of "High Resistance to Antibiotics Recommended in Standard Treatment Guidelines in Ghana: A Cross-Sectional Study of Antimicrobial Resistance Patterns in Patients with Urinary Tract Infections between 2017–2021"

_ijerph, 2022, doi:10.3390/ijerph192416556_

Round 1

Reviewer 1 Report

 Dear editor of IJERPH

I read carefully the manuscript titled” High resistance to antibiotics recommended in standard treatment guidelines in Ghana: A cross-sectional study of antimicrobial resistance patterns in patients with urinary tract infections between 2017 -2021”. The paper is written well in a scientific way and has new findings. I recommend the manuscript for publication with following minor changes.

·         It’s better to summarize the “General Setting” and “Specific setting” section.

·         Abbreviations must be fully spelled only out at the first time.

·         Please mention the including criteria for the UTI samples.

·         Conclusion should be written based on your results instead of overall statement

·         I suggest citation of following related published papers on the topic to extend both introduction and discussion.

 -          Halaji M et al. Phylogenetic Group Distribution of Uropathogenic Escherichia coli and Related Antimicrobial Resistance Pattern: A Meta-Analysis and Systematic Review.  Front Cell Infect Microbiol. 2022 Feb 25;12:790184. doi: 10.3389/fcimb.2022.790184. eCollection 2022.

-            Raeispour M et al. Antibiotic resistance, virulence factors and genotyping of Uropathogenic Escherichia coli strains.Antimicrob Resist Infect Control. 2018 Oct 3;7:118. doi: 10.1186/s13756-018-0411-4. eCollection 2018.

Reviewer 2 Report

In this study, the authors describe the trends in antimicrobial resistance of uropathogens isolated from the largest private sector laboratory in Ghana over five years. They reviewed positive urine cultures at the MDS Lancet Laboratories from 2017 to 2021. This is a well-written manuscript, however, to strengthen it, the authors should address the following concerns.

Major Comments

1.      The authors majorly relied on the phenotypic analysis to determine antibiotic residue present in 1,986 (9.9%) culture-positive samples and reported their findings. Tables 3 and 4 show the prevalence of antimicrobial resistance to common antibiotics used in treating UTIs, all these results were based on the phenotypic analysis. To strengthen this study, the authors must do genotypic assays to confirm what was observed in Tables 3 and 4, two to three isolates from each organism can be used. Normal PCR targeting specific Antibiotic resistance genes (ARGs) should be performed.

The authors should note that Antibiotic resistance genes (ARGs) have accelerated microbial threats to human health in the last decade. Many genes can confer resistance, this offers a strong basis for genotypic assays. Factors such as the abundance, propensity for lateral transmission, and ability of ARGs to be expressed in pathogens are all important.

2.      In this kind of study, we would expect a pattern from isolates like Resistant (R), Intermediate (I), or susceptible (S). As indicated in the title “High resistance to antibiotics recommended in standard treatment guidelines in Ghana: A cross-sectional study of antimicrobial resistance patterns in patients with urinary tract infections between 2017 -2021”. The authors should explain why they only reported Resistant (R) and not Intermediate (I) as well as susceptible (S) yet they analyzed positive urine cultures.

Minor comments/suggestions

Line 26. Consider writing ‘ESBL’ in full as it’s the first time introduced to the readers.

Line 128: put a full stop after “……… blood agar and antimicrobial plates [19]”.

Line 134: put a full stop after “……… or Microscan [20]”.

Line 135: put a full stop after “……… on Mueller Hinton agar [21]”.

Organisms' names should be in italic, ensure this is corrected.

Typos and grammatical errors should be checked in the entire manuscript.
